# Income Disparity and Mental Wellbeing among Adults in Semi-Urban and Rural Areas in Malaysia: The Mediating Role of Social Capital

**DOI:** 10.3390/ijerph19116604

**Published:** 2022-05-28

**Authors:** Mas Ayu Said, Govindamal Thangiah, Hazreen Abdul Majid, Rozmi Ismail, Tan Maw Pin, Hussein Rizal, Mohd Azlan Shah Zaidi, Daniel Reidpath, Tin Tin Su

**Affiliations:** 1Department of Social and Preventive Medicine, Faculty of Medicine, University of Malaya, Kuala Lumpur 50603, Malaysia; mas@ummc.edu.my (M.A.S.); govindamal.t@gmail.com (G.T.); hazreen@ummc.edu.my (H.A.M.); husseinriz@um.edu.my (H.R.); 2Centre for Epidemiology and Evidence-Based Practice, Department of Social and Preventive Medicine, Faculty of Medicine, University of Malaya, Kuala Lumpur 50603, Malaysia; 3South East Asia Community Observatory (SEACO) & Global Public Health, Jeffrey Cheah School of Medicine and Health Sciences, Monash University Malaysia, Subang Jaya 47500, Malaysia; daniel.reidpath@icddrb.org; 4Department of Nutrition, Faculty of Public Health, Universitas Airlangga, Surabaya 60115, Indonesia; 5Centre for Population Health, Department of Social and Preventive Medicine, Faculty of Medicine, University of Malaya, Kuala Lumpur 50603, Malaysia; 6Psychology and Human Wellbeing Research Centre, Faculty of Social Sciences and Humanities, Universiti Kebangsaan Malaysia, Bangi 43600, Malaysia; rozmi@ukm.edu.my; 7Department of Medicine, Faculty of Medicine, University of Malaya, Kuala Lumpur 50603, Malaysia; mptan@ummc.edu.my; 8Center for Sustainable and Inclusive Development Studies, Faculty of Economics and Management, Universiti Kebangsaan Malaysia, Bangi 43600, Malaysia; azlan@ukm.edu.my; 9icddr,b, International Centre for Diarrhoeal Disease Research, Dhaka 1212, Bangladesh

**Keywords:** income, mental health, social capital, mediation, older adults

## Abstract

Mental illness is rising worldwide and is more prevalent among the older population. Among others, socioeconomic status, particularly income, has a bearing on the prevalence of mental health. However, little is known about the underlying mechanism that explains the association between income and mental health. Hence, this study seeks to examine the mediating effect of social capital on the association between income and mental illness. Cross-sectional data consisting of 6651 respondents aged 55 years and above were used in this study. A validated tool known as the Depression, Anxiety and Stress Scale, 21 items (DASS-21) was applied to examine mental illness, namely depression, anxiety, and stress. The Karlson, Holm, and Breen (KHB) method was employed to assess the intervening role of social capital on the association between income and mental illness. Results showed that those who disagreed in trust within the community had the highest partial mediation percentage. Those who disagreed in reciprocity, however, had the lowest partial mediation percentage, which explained the positive association between the middle 40% (M40) of the income group and depression, anxiety, and stress. Overall, the study suggests the need to increase trust and attachment within society to curb the occurrence of depression and anxiety.

## 1. Introduction

Mental illness is a growing global health crisis. It was ranked as the sixth highest contributor to disability-adjusted life years in 2017 [1]. Psychological distress such as major depressive disorder and generalized anxiety disorders were among the top 20 global burdens of disease and leading causes of years lived with a disability (YLDs) in 2013 [2]. This trend has only exacerbated since the onset of the COVID-19 pandemic, as the global prevalence of anxiety and depression increased by a massive 25 percent since the first year of the pandemic [3]. Individuals with mental illness also face higher rates of physical abuse and mortality than those without. For instance, the odds of all-cause mortality are 1.7-times greater for depressed individuals than the general population [4]. Previous studies have established that, along with biological factors, social determinants also affect individual mental health [5,6,7,8].

Using data from the Health and Retirement Study, [9] evidenced a positive impact of high income on mental health by attenuating the psychosocial stress related to financial hardship. This is consistent with the findings of a study in China, which showed that high income is associated with better mental health status due to better life satisfaction, living conditions, and accessibility to health care [10]. Similarly, another study in China showed that large savings are associated with low depression levels [11]. A longitudinal study also showed that individuals with low annual household income are more likely affected by mental disorders such as anxiety than their counterparts [12]. Corroborating this, findings from several other studies also showed the same link between income and mental disorders [13,14].

However, the social selection hypothesis proposed that mental illnesses are more predisposed to low socioeconomic status (SES), owing to genetic factors, hospitalization, and loss of work [12]. In line with this, past studies showed that those diagnosed with depression or anxiety are linked with low income levels vis-à-vis those who are undiagnosed with depression or anxiety [15,16]. It was previously established that income inequality affects mental health through the material and psychosocial pathway. On one hand, the psychosocial pathway suggests that status comparison leads to poor social cohesion and low trust in the community, which causes stress. Consequently, this impacts individual’s self-esteem and their health-related behaviors.

On the other hand, the material pathway proposed that poverty and deprivation are linked to high stress level, especially in societies with high inequality. These underlying mechanisms could also be explained by the status anxiety, social capital, and neo-materialist hypothesis [17,18,19,20]. While the status anxiety hypothesis proposes that unequal status order causes negative feelings such as shame and distrust, which affects mental health status, the neo-materialist hypothesis posits that variation in the availability and use of social infrastructure influences mental health [20].

In addition, the social capital hypothesis suggests that high income inequality causes status inequality, poor social cohesion, and low trust, which in turn affects health-related behaviors, access to health services and facilities, as well as psychosocial processes [20]. Social capital is a dynamic process that facilitates social integration among individuals within a community that seeks to achieve social goals [7,20]. Widening income inequality in communities erodes social capital and trustworthiness within societies through status comparison [20,21], thus creating loneliness and isolation [7].

Social participation is among the key criteria of successful aging for older adults who are often at risk of social isolation. Most older populations experience a lack of interaction with others as they age, resulting in low social capital values. With inadequate social support, the older population most likely faces depression [22]. A study on rural areas of Korea showed a positive link between low social connectedness and depression among older adults [23]. A follow-up study in the United Kingdom also showed that inconsistent social contact increases the risk of isolation and depression among the older population [24]. Consistently, several other studies also showed that high social capital among older adults in rural areas results in better mental health status [25,26].

Moreover, older adults who live in areas with greater income inequality, especially in rural areas [27], have a high risk of developing depression [28]. Income inequality also affects the social cohesion among community members [29], which in turn impacts their mental health status [20]. The paucity of studies investigating the underlying mechanism of social capital on the association between income and mental health prompts this study to probe the mediating effect of social capital on the individual level. Hence, this study seeks to examine the mediating effect of social capital on the association between income and mental illness, specifically among those age 55 years and above living in semi-urban and rural areas of Malaysia.

## 2. Materials and Methods

The South East Asia Community Observatory (SEACO) team located at the district of Segamat, in the state of Johor, Malaysia collected the sample of this study [30]. Segamat and its five sub-districts were chosen based on the strong pre-existing relationship between the Jeffery Cheah School of Medicine and Health Sciences (JCSMHS) and the district, as well as the state health administration, which is essential to conduct this research [30]. Segamat has a marked ethnic breakdown that closely reflects the national proportions of Malays (60%), Chinese (23%), and Indians (7%), as well as an equal gender composition (male: 49%; female: 51%) [30].

### 2.1. Study Design

This study used a cross-sectional study design. A total sample of 25,512 respondents enrolled in the SEACO health survey conducted in 2013. Of this number, 6651 pieces of complete information on social capital variables were available for participants aged only above 55 years. Hence, this sample was used in the analysis of this study. All trained enumerators and staff briefed participants about the objectives of the survey conducted. Only participants who gave written consent were recruited and enrolled in this study. Respondents were approached at their respective residence to gather information on their sociodemographic background (age, gender, education, employment status, income, marital status, ethnicity), social capital, and mental health status using standardized health data collection tools. This information was recorded directly into Android mobile devices and tablets with survey forms designed in Open Data Kit (ODK). Data recorded on the tablets were then encrypted and uploaded to a secure server. The study was conducted in accordance with the Declaration of Helsinki and was approved by the Monash University Human Research Ethics Committee (2013-3837-3646).

### 2.2. Study Instrument

Mental illnesses such as depression, anxiety, and stress were used as dependent variables in this study. Depression, anxiety, and stress were assessed using the Depression, Anxiety and Stress Scale, 21 items (DASS-21). These outcome variables were further defined as dichotomous variables (0,1), where 0 is categorized as those who do not experience depression, anxiety, and stress while 1 classifies those who have mild, moderate, severe, and extremely severe depression, anxiety, and stress levels. A validated Malay version of the DASS-21 self-reported questionnaire, which was confirmed reliable and effective among Malaysians, was used to evaluate depression, anxiety, and stress [31]. The Cronbach’s alpha value for overall items was very good 0.90. For the depression, anxiety, and stress scales, the values were 0.84, 0.74, and 0.79, respectively [31].

### 2.3. Explanatory Variables

#### Monthly Household Income

Monthly household income groups were characterized into two categories, namely bottom 40% (B40) and middle 40% (M40), based on the income thresholds provided by the Department of Statistics Malaysia (DOSM) in 2014. There were no respondents that belonged to the top 20% (T20) income group. The B40 income group was defined as individuals with monthly household incomes below RM3860, while the M40 income group was those with a monthly household income between RM3860 and RM8319. The income category was represented in binary, whereby 0 is defined as B40 and 1 is defined as M40. Income distributions are often skewed, and they are apparent in many past works. Therefore, the use of the mean as a measure of income will not be reliable because it loses its power to produce accurate results [32]. In addition, previous studies that used mean income often ignored the different impacts of demographic and socioeconomic variables on each income level [33]. Hence, income groups that consider the income distribution are better and more commonly used to examine income differentials.

### 2.4. Mediators

#### 2.4.1. Individual Social Capital Measure

There are two distinctive notions of social capital in the literature. On one hand, Putnam considers social capital as a collective attribute that features the trust, norms, and networks within an organization [34]. On the other hand, Bordieu conceptualizes social capital as resources available to individuals within their network [34]. Hence, it is specified into two different categories, individual and aggregated levels [35]. In this study, we focused on the individual level of social capital, as it helped to avoid interpretation problems arising from measurement issues of aggregated data [36].

Moreover, since individuals are generally more involved in decisions to invest in social capital than communities, it is better to assess it at an individual level [37]. Since there is an absence on an agreed standardized measure of social capital, this study operationalized it in a way analogous to Kawachi, Kennedy [38], which focused on trust and reciprocity. Social capital is differentiated into four items, namely, reciprocity or “If I do nice things for someone, I can anticipate that they will respect and treat me just as well as I treat them”, cooperation among community members or “If I see people who cooperate with each other, I also feel that I would help someone in need”, trustworthiness of community or “In a difficult situation, I can count on the help from people in my local community members”, and attachment to local community members or “I feel a strong attachment to my local community”. The Cronbach’s alpha was 0.85 for all four items.

All the social capital variables were categorized as binary variables (0,1), where 1 is defined as those who totally disagree or disagree or neither agree nor disagree in reciprocity, cooperation, trust, and attachment to local community, while 0 is classified as those who totally agree or agree in reciprocity, cooperation among community, trust in community, and attachment to local community.

#### 2.4.2. Control Variables

Sociodemographic variables such as age, gender, marital status, ethnicity, educational level, and employment status were controlled for in the analysis to avoid the potential confounding effects of these variables on the association between income, social capital, and mental health status.

### 2.5. Statistical Analysis

The frequency of variables was recorded, and a Chi-Squared test was conducted to identify the presence of significant bivariate associations between the predictor variables and the depression, stress, and anxiety levels. Next, a mediation analysis was applied to investigate the mediating effect of individual social capital measures on the association between income groups and mental health status. A robust Huber-White sandwich estimator was also used to avoid heteroskedasticity issues, further contributing to the rigorousness of the model produced. All analyses were performed using STATA version 14.0 StataCorp, 2015, with a five percent and ten percent significance level.

A multiple path mediation model was deployed to assess the aim of this study. In doing so, individual social capital measures were used as the mediators in the relationship between income and depression, anxiety, and stress. This was examined controlling for the age, gender, ethnicity, marital status, education, and employment status of participants. The mediation framework was as shown in Figure 1.

Since the dependent variable was dichotomous, a binary logistic regression was chosen in this study. With this, the linear effect between the independent and mediator variables on the outcome were captured. In doing so, the size of effect and scaling of parameters were affected. The Karlson, Holm, and Breen (KHB) method [39,40] was applied to conduct the mediation analysis, which decomposed the variables to adjust to the framework of nonlinear probability models (binary logistic regression). The rescaling problem was addressed by incorporating the standardized residuals of the regressor X (independent variables) on Z (mediators) in the reduced model to ensure that the estimated coefficients in the models were measured using the same scale, so that the comparison of coefficients was standardized across the regression models to eliminate the effects of conflation [39].

The mediation method was adopted because it helped explain the focal association that exists between income levels and the risk of depression, stress, and anxiety. There are two types of mediation effects, namely partial mediation and full mediation. The former arises when the *p*-value of the total, direct, and indirect effects are significant. The latter occurs when the *p*-value of direct effects are insignificant, while the *p*-value of the total and indirect effects are significant.

## 3. Results

Table 1 below presents the prevalence and descriptive statistics of the variables included in the study. The average age of the respondents in the sample was 65 years, with a standard deviation of 7.7 (mean ± SD = 65.5 ± 7.7). The sample was largely comprised of Malays (61.8%), married couples (75.2%), and B40 income groups (91.1%), which closely represented the characteristics of a semi-urban and rural population.

Table 1 also records the low shares of respondents who did suffer from self-reported depression (16.5%), stress (6.9%), and anxiety (22.4%) levels. In addition, the bivariate associations showed significant results between the demographic variables (marital status and ethnicity), socioeconomic status (income, education, employment status), individual social capital variables, and the dichotomous levels of depression, stress, and anxiety (Appendix A).

In addition, the characteristics of individuals without stress, depression, and anxiety levels included those who were widows, with other education such as vocational schools and more, were self-employed, B40, agreed with reciprocity and cooperation, and those who disagreed or were neutral in trust within the community and attachment to local community.

Individuals with mild, moderate, severe, and extremely severe stress, anxiety, and depression levels consisted of individuals who were single, had secondary educated, had under paid employment, were from the M40 income group, disagreed or were neutral in reciprocity and cooperation, and agreed with trust within the community and attachment to local community.

### Mediation Analysis

The results of the mediation analyses are shown in Table 2. A partial mediation was achieved for the M40 income group, as the *p*-values of the total and direct effects of the M40 income group on anxiety, stress, and depression were significant (*p* < 0.05) compared to the B40 income group. The partial mediating effects of those who disagreed or were neutral in the trust within local community members were consistently the strongest between the M40 income group and anxiety (30.8%) and depression (24.8%). However, those who disagreed or were neutral in reciprocity had the lowest partial mediation rate (Reciprocity_anxiety_ = 22.8%; Reciprocity_depression_ = 18.5%) on these relationships, adjusting for socio-demographic variables in this study.

Similarly, those who disagreed or were neutral in reciprocity also had the lowest mediation rate (12.0%) between the M40 income group and stress level. However, those who disagreed or were neutral in attachment to local community had the highest partial mediation percentage (19.7%) between the M40 income group and stress level. The partial mediation showed that the M40 income group had an attenuated effect on anxiety, stress, and depression once the mediators were included. The total and direct effect of the M40 income group on anxiety, depression, and stress was positive and significant for all models. The mediation effects of reciprocity, trust in community, and attachment to local community on the association between income and anxiety, depression, and stress were all present and significant (*p* < 0.05).

The positive impact of the total and direct effects of the M40 income group compared to the B40 income group on anxiety and depression was explained by the positive indirect effect of reciprocity, trust in community, and attachment to local community. However, the mediation effect of the cooperation among community members on the association between income and anxiety and depression was insignificant (indirect effect of cooperation_anxiety_ = 0.15; indirect effect of cooperation_depression_ = 0.17), except for its association with the link between income and stress (indirect effect of cooperation_stress_ = 0.17).

## 4. Discussion

The evidence produced in the present study show the significance of raising awareness on the importance of inculcating reciprocity, trust in community, cooperation, and attachment to local community members. The lack of support on social capital values among the M40 population engendered social segregation, which explained their lack of interaction with other members of the community. Thus, this elucidated the positive association between the M40 group and depression, anxiety, as well as stress. The study also showed that the B40 group had a better mental health status, owing to a greater social connectedness among themselves in this study.

This concurred with the findings of a study in Malaysia, which showed that social capital was high among the B40 low income group that gave rise to better mental health status [41]. Another study in the urban areas of the East coast of the Peninsula Malaysia showed that high social capital and good mental health were predominant among low income earners [42]. One explanation was that local initiatives, such as PeKa B40, mySalam, and household living aid (BSH), are continuously improved and disseminated among this group [43], more so than the M40 income group. Many of these programs are facilitated by local leaders and volunteers among these communities. This provides a platform for the continuous improvement of social capital within this population group.

Among the four individual social capital measures, those who disagreed or were neutral in their trust within their local community had the highest mediation percentage, which explained the underlying positive association between the M40 group and depression and anxiety, controlling for demographic and SES variables. This partly concurred with the outcome of a study in South Africa, which showed that lower social trust is associated with higher depression levels [44]. Similarly, findings from South Korea also showed that low interpersonal trust is positively associated with depression [45]. With respect to the association between income and mental health problems, a study in the United States, however, showed that those with low income are more at risk of mental illness than those with high income [46]. Interestingly, in Japan, an inconsistent association between income and depression was reported [17].

Results from this study also showed that those who disagreed or were neutral with reciprocity had the lowest partial mediation percentage that explicated the positive association between M40 and depression, anxiety, and stress, adjusting for demographic and SES variables. Nonetheless, a study in Japan showed that reciprocity did not modify the association between income and depression [47]. A study in South Korea showed that reciprocity was associated with new-onset depression [45]. However, another study showed that neighborhood social reciprocity is associated with better mental health status [48]. Another study also showed that the older community, who receive pensions, could maintain reciprocity, which in turn results in good mental health status [49]. Apart from that, attachment to local community also partially mediates the positive association between income and anxiety, depression, and stress. Using data from the Hamilton Household Quality of Life survey, a study showed that those with a greater sense of attachment to their neighborhood have a better mental health status [50].

Several intervention programs that aim to uplift social capital values were carried out at the individual level, community level, and a combination of both. While individual level interventions include the exchange of resources between members of a social network [51,52], community level interventions are performed to help groups organize activities and act collectively. For instance, home visitation health programs conducted for mothers with newborns within the community are based on social trust and a sense of security, where they are used as a mediator to attenuate stress levels among mothers of newborns [53]. Moreover, visiting and communicating with the elderly is another form of social capital intervention that eradicates loneliness among them at an intrapersonal level [54]. Thus, as intervening variables, the social capital variables can achieve the desired outcome. Hence, this shows the need to strengthen social capital variables, such as trust in the community, to reduce mental illness among the M40 community, as proposed in this study.

The strengths of this study include the availability of a large sample size, which helped to produce accurate findings that are generalizable to the older community in the rural and semi-urban areas of Segamat. The study also had an ethnic breakdown that closely represented the national proportion of ethnicity. However, this study consisted of some limitations that should be surmounted. These include the lack of a longitudinal assessment on the association between social capital, income, and mental health, which sheds light on the causal explanation. In addition, the dependent variable was dichotomized, as for those who do not experience depression, anxiety, and stress; and those who have mild, moderate, severe, and extremely severe depression, anxiety, and stress levels. In this way, it was not possible to distinguish between the different types of discomfort people experience. Currently, there are no normative scores for the DASS-21 in Malaysia; however, two validation studies were conducted but were not derived from representative populations [55,56], which may have led to measurement bias. Finally, the study was conducted in 2013 and did not take into account the massive effect the COVID-19 pandemic has had on mental illness among the Malaysian population. There are changes in terms of the social interaction and mental health of respondents due to the pandemic that might affect the association between income and mental illness [57].

## 5. Conclusions

Overall, the findings of this study suggest the necessity to increase awareness of the importance of social capital values, especially trust in the local community, among the M40 older community to achieve good mental health status. The M40 community who were either previously in the Top 20 or Bottom 40 population could have also faced a social status comparison, which might explain their lack of trust within the community, thus causing depression or anxiety. However, most Malaysian rural folks, especially the B40 population, enjoy better social capital such as trust, attachment to local community, and cooperation among community members without expecting anything in return, which could be a social norm among them. Several strategies, such as relational community engagement programs, which help to create opportunities for local community members to talk about trust and expectations, could be implemented to elevate the level of trustworthiness within the M40 community. Apart from that, local community members could also hold meetings to resolve conflict between each other, thus enhancing the level of trustworthiness within the community. Future studies should evaluate the trust in local community as a potentially modifiable risk factor. Moreover, the outcome from this study informs multiple stakeholders to execute proper intervention programs that would improve the social capital values among the M40 population, such as those already implemented among the B40 income group.

## Figures and Tables

**Figure 1 ijerph-19-06604-f001:**
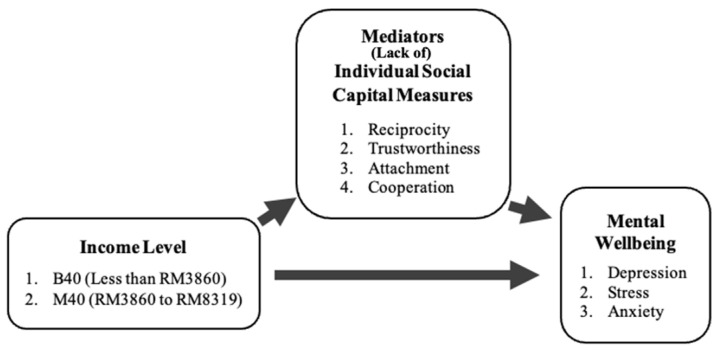
Framework of the mediation analysis.

**Table 1 ijerph-19-06604-t001:** Sample characteristics of the semi-urban and rural community (*n* = 6651).

Characteristics	Prevalence, *n* (%)	Mean ± SD
Age (*n* = 6651)	6651 (100.0)	65.5 ± 7.7
Gender (*n* = 6651)		
Female	3556 (53.5)	
Male	3095 (46.5)	
Ethnicity (*n* = 6646)		
Malays	4111 (61.9)	
Indians	522 (7.9)	
Chinese	1957 (29.4)	
Others	56 (0.8)	
Marital Status (*n* = 6649)		
Single	167 (2.5)	
Married	5000 (75.2)	
Separated/divorced/other	105 (1.6)	
Widow	1377 (20.7)	
Education Level (*n* = 6487)		
None	381 (5.9)	
Primary	3822 (58.9)	
Secondary	1804 (27.8)	
Tertiary	150 (2.3)	
Others	330 (5.1)	
Employment Status (*n* = 6617)		
Housewife	2163 (32.7)	
Unemployed	1420 (21.5)	
Paid employment	901 (13.6)	
Pensioners	907 (13.7)	
Self employed	1226 (18.5)	
Income groups (*n* = 6145)		
B40 (Less than RM3860)	6061 (99.7)	
M40 (RM3860 to RM8319)	84 (0.3)	
Individual social capital variables		
Reciprocity (*n* = 6542)		
Agree	4439 (67.9)	
Disagree or “neutral”	2103 (32.1)	
Cooperation (*n* = 6549)		
Agree	4780 (73.0)	
Disagree or “neutral”	1769 (27.0)	
Trust in community (*n* = 6527)		
Disagree or “neutral”	4290 (65.7)	
Agree	2237 (34.3)	
Attached to community (*n* = 6534)		
Disagree or “neutral”	4176 (63.9)	
Agree	2358 (36.1)	
Mental health (DASS-21)		12.39 (17.80)
Depression (*n* = 6601)		
Normal	5511 (83.5)	
Mild and above	1090 (16.5)	
Anxiety (*n* = 6606)		
Normal	5128 (77.6)	
Mild and above	1478 (22.4)	
Stress (*n* = 6591)		
Normal	6135 (93.1)	
Mild and above	456 (6.9)	

Note: SD—standard deviation.

**Table 2 ijerph-19-06604-t002:** Mediating effects of social capital on the association between income groups and anxiety, depression, and stress.

Outcome	Mediator			
	Reciprocity	Cooperation among Community Members	Trust in Community	Attached to Local Community
Anxiety				
Total effect M40	0.74 *** (0.25)	0.79 *** (0.27)	0.78 *** (0.27)	0.78 *** (0.27)
Direct effect M40	0.57 ** (0.25)	0.63 ** (0.27)	0.54 * (0.27)	0.57 ** (0.27)
Indirect effect	0.17 ** (0.08)	0.15 (0.10)	0.24 ** (0.10)	0.21 ** (0.10)
Mediation (%)	22.8	19.5	30.8	26.5
Partial mediation	yes	no	yes	yes
Full mediation	no	no	no	no
Depression				
Total effect M40	1.01 *** (0.26)	1.09 *** (0.28)	1.07 *** (0.28)	1.08 *** (0.29)
Direct effect M40	0.82 *** (0.26)	0.92 *** (0.28)	0.80 *** (0.28)	0.82 *** (0.29)
Indirect effect	0.19 ** (0.08)	0.17 (0.11)	0.26 ** (0.11)	0.24 ** (0.11)
Mediation (%)	18.5	15.3	24.8	22.6
Partial mediation	yes	no	yes	yes
Full mediation	no	no	no	no
Stress				
Total effect M40	1.28 *** (0.28)	1.34 *** (0.29)	1.34 *** (0.31)	1.31 *** (0.30)
Direct effect M40	1.12 *** (0.28)	1.17 *** (0.29)	1.08 *** (0.31)	1.05 *** (0.30)
Indirect effect	0.15 ** (0.07)	0.17 ** (0.10)	0.26 ** (0.12)	0.26 ** (0.11)
Mediation (%)	12.0	12.9	19.5	19.7
Partial mediation	yes	yes	yes	yes
Full mediation	no	no	no	no

Note: ***, **, * Significant at 1%, 5% and 10% significance level; B- coefficient, SE- standard error; Model is adjusted for ethnicity, marital status, age, gender, education and employment status.

## Data Availability

The data presented in this study are openly available in figshare at doi:10.6084/m9.figshare.19572292 (accessed on 24 May 2022).

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
