# Peer review of "Income Disparity and Mental Wellbeing among Adults in Semi-Urban and Rural Areas in Malaysia: The Mediating Role of Social Capital"

_ijerph, 2022, doi:10.3390/ijerph19116604_

Round 1

Reviewer 1 Report

Some comments are suggested:

  • It would be interesting if the keywords are DeCS/MeSH descriptors.
  • In the introduction, it is important to update the information on epidemiological data, since they are provided from 2013 or earlier. Also taking into account that we have subsequently had a pandemic, which affects mental health, it should be better contextualized.
  • In reference to this paragraph "On the contrary, a study in Japan recorded an inconsistent association between income and depression [17]. Hence, this suggest that there exists a bidirectional association between income and mental health status and the relationship is inconclusive. This also further indicates that there could be underlying mechanisms that explain the association between income and mental health" it is necessary to avoid making conjectures without scientific support and even less, in the introduction of the manuscript.
  • In the introduction, although social capital is explained in material and method, it is important that they indicate that they will focus on the individual level proposed by Bourdieu
  • In addition, the last paragraph of the introduction must clearly contain the objective of the study.
  • The material and methods section must begin with the definition of the type of design to be studied. And it is from there, where the field of study is explained.
  • In the instrument used, validation data such as Cronbach's alpha must be included.
  • In the results, the relative frequencies must be accompanied by the n
  • Data is being repeated in written form identically to what is expressed in the tables. This supposes a repetition that must be eliminated (or in the table or in the text or in the text a more clarifying grouping that provides different information)
  • The paragraphs of the results are very long, which makes it difficult to understand them. It is recommended that more points and paragraphs be used and the information be summarized, even more so if they are repeated with the tables.
  • It is recommended that the conclusions be more specific and implications for social, political and socio-health practice are established.

Author Response

1

It would be interesting if the keywords are DeCS/MeSH descriptors.

-

-

2

In the introduction, it is important to update the information on epidemiological data since they are provided from 2013 or earlier. Also taking into account that we have subsequently had a pandemic, which affects mental health, it should be better contextualized.

Added:

“This trend has only exacerbated since the onset of the COVID-19 pandemic as the global prevalence of anxiety and depression increased by a massive 25 percent since the first year of the pandemic [4].”

2 (50-52)

3

In reference to this paragraph "On the contrary, a study in Japan recorded an inconsistent association between income and depression [17]. Hence, this suggest that there exists a bidirectional association between income and mental health status and the relationship is inconclusive. This also further indicates that there could be underlying mechanisms that explain the association between income and mental health" it is necessary to avoid making conjectures without scientific support and even less, in the introduction of the manuscript.

Removed

2 (69-73)

4

In the introduction, although social capital is explained in material and method, it is important that they indicate that they will focus on the individual level proposed by Bourdieu

Added and revised sentence:

“The paucity of studies investigating the underlying mechanism of social capital on the association between income and mental health prompts this study to probe the mediating effect of social capital on the individual level.”

3 (114)

5

In addition, the last paragraph of the introduction must clearly contain the objective of the study.

Added objective:

“Hence, this study seeks to examine the mediating effect of social capital on the association between income and mental illness specifically among those age 55 years and above living in semi-urban and rural areas of Malaysia.”

3 (114-117)

6

The material and methods section must begin with the definition of the type of design to be studied. And it is from there, where the field of study is explained.

Added:

“This study used a cross-sectional study design.”

3 (129)

7

In the instrument used, validation data such as Cronbach's alpha must be included.

Cronbach’s alpha value for overall items was very good 0.90. For depression, anxiety and stress scales the values were 0.84, 0.74 and 0.79 respectively [31].

The Cronbach’s alpha is 0.85 for all four items.

3 (152-154)

4 (195)

8

In the results, the relative frequencies must be accompanied by the n.

Added relative n frequency to table 1

6-7 (268-269)

9

Data is being repeated in written form identically to what is expressed in the tables. This supposes a repetition that must be eliminated (or in the table or in the text or in the text a more clarifying grouping that provides different information)

Revised results section:

·       Combined tables 2, 3 and 4

·       Eliminate repetition

8 (341-344)

10

The paragraphs of the results are very long, which makes it difficult to understand them. It is recommended that more points and paragraphs be used and the information be summarized, even more so if they are repeated with the tables.

Revised and reduced clutter in the result section. Structured paragraphs to improve readability.

Kindly refer track changes in the main document for full changelog.

6-7 (244-297)

11

It is recommended that the conclusions be more specific and implications for social, political and socio-health practice are established.

Added in discussion and revised conclusion:

“One explanation is that local initiatives such as PeKa B40, mySalam and household living aid (BSH) are continuously improved and disseminated among this group [42], more so than the M40 income group. Many of these programs are facilitated by local leaders and volunteers among these community. This provides a platform for continuous improvement of social capital within this population group.”

“Moreover, outcome from this study informs multiple stakeholders execute proper intervention programs which would improve the social capital values among the M40 population such as those already implemented among the B40 income group.”

Note: PeKa B40, mySalam and household living aid (BSH) are economic and health aids provided to the B40 income group by the government.

8 (358-362)

10 (466-469)

Reviewer 2 Report

The authors' work is worthwhile in its way, above all for having measured on a large sample of subjects that are sometimes difficult to reach. Nonetheless, I have several doubts that I hope the authors will be able to address through a proper revision. 

First of all, I really would like to understand the need to dichotomize social capital and DASS. Given your sample size, statistical power is not an issue, so why dichotomize? Most of all, why dichotomize DASS including in the same category mild, moderate, severe, and extremely severe depression, anxiety, and stress levels? In this way, the authors are not able to distinguish between the different types of "discomfort" people experience. 

Moreover, I would appreciate seeing the score of the DASS you collected (mean and s.d) together with normative scores for Malaysia. Deviations from the normative score are important to be highlighted in your specific case. 

It would be important to report descriptive statistics and results in a sex-sensitive way (i.e., disaggregating data and results by sex if applicable). Thus, it would be important to re-run the models and see if differences will emerge considering males and females separately. Indeed, sex, as the authors surely know, is strongly related to DASS variables. 

Author Response

No

Comments

Changes Made

Page (Line)

1

I really would like to understand the need to dichotomize social capital and DASS. Given your sample size, statistical power is not an issue, so why dichotomize? Most of all, why dichotomize DASS including in the same category mild, moderate, severe, and extremely severe depression, anxiety, and stress levels? In this way, the authors are not able to distinguish between the different types of "discomfort" people experience. 

We dichotomize social capital and DASS to suit our objective of the study which is to examine the role of social capital (lack of) in mediating the association between income disparity and mental illness (having depression, anxiety, or stress).  Specifically, we look at how being lack of social capital can have an effect on the mental illness of the respondents. Due to the lack of sample in one of the groups, the outcome was dichotomize. We acknowledge that dichotomizing our data may lead to loss of granularity of information, and had prior to this done this to aid interpretation.

-

2

Moreover, I would appreciate seeing the score of the DASS you collected (mean and s.d) together with normative scores for Malaysia. Deviations from the normative score are important to be highlighted in your specific case.

Currently there is no normative scores for DASS-21 in Malaysia as far as the authors are aware. However, there are two validation papers conducted in Malaysia, but were not derived from representative populations:

·       https://www.aseanjournalofpsychiatry.org/articles/concurrent-validity-of-the-depression-and-anxiety-components-in-the-bahasa-malaysia-version-of-the-depression-anxiety-an.pdf

·       http://www2.psy.unsw.edu.au/DASS/Malaysian/ramli_Bahasa_article.pdf

A representative outlook of the mental health in Malaysia can be seen via the National Health Morbidity Survey (NHMS) in 2015 which shows the prevalence of mental problems among adults in Malaysia was 10.7% (1996) and this has increased to 29.2% as referenced below:

·       https://www.moh.gov.my/moh/resources/nhmsreport2015vol2.pdf

·       https://www.moh.gov.my/moh/resources/Penerbitan/Laporan/Umum/Mental%20Healthcare%20Performance%20Report%202016.pdf

The mean (SD) of DASS-21 is added to Table 1:

The mean (SD) is 12.39 (17.80) / median (IQR) = 4.0 (18.0).

7 (270)

3

It would be important to report descriptive statistics and results in a sex-sensitive way (i.e., disaggregating data and results by sex if applicable). Thus, it would be important to re-run the models and see if differences will emerge considering males and females separately. Indeed, sex, as the authors surely know, is strongly related to DASS variables.

The present study aims to examine the mediating effect of social capital on income disparity and mental condition associations. As a reanalysis of this model would fall short of the scope of this study, it’ll be more suitable for a future paper. In addition, Table A1 in the appendix has also indicated the bilateral associations of gender with the DASS-21 variables. Regardless, we are happy to explore the model analysis and add it to the manuscript if requested by the reviewer.

11-13 (509-510)

Reviewer 3 Report

The paper analyzes the impact of social capital as a mediating variable to explain the association between income and mental illness. Using a 2013 data of 6651 respondents in Malaysia, results suggest the necessity to increase the trust and attachment within a society in curbing mental illness, particularly depression and anxiety. 

In general, the paper is well-written and proposes a good academic contribution. However, its international significance and discussion in a broader perspective should be improved. Specifically,

  1. The study is too focused in Malaysia. The Title must be changed to "Income Disparity and Mental Wellbeing among Adults in Semi-urban and Rural Areas in Malaysia: The Mediating Role of Social
    Capital", otherwise, s
    1. Explain why Malaysia was selected as a case study.
    2. Discuss the main findings with an international perspective.
  2. The data was taken from previous survey in 2013. This should be discussed as one of limitations of the study. First, there might be changes in economic status in the case country a decade ago and now. Second, in the last two years, there are changes in terms of social interaction and mental health of respondents due to the pandemic, that might affect the association between income and mental illness. See for instance <https://doi.org/10.3390/ijerph17217947>, <https://doi.org/10.3390/bs11050064>, and <https://doi.org/10.3390/ijerph19063158>.
  3. The findings should be discussed in a broader perspective, for instance, (economic and/or health) policy implications.  
  4. The references must be improve by citing the most recent (2020 or newer) and relevant studies.
  5. Figures and tables should be self-explanatory and stand-alone. Define all acronyms and uncommon terms (e.g. M40, B40, RM3860,...) 
  6. Minor issues on capitalization and consistency in using comma for numbers (e.g. 25,512 respondents vs 6651).

Author Response

No

Comments

Changes made

Page (line)

1

The study is too focused in Malaysia. The Title must be changed to "Income Disparity and Mental Wellbeing among Adults in Semi-urban and Rural Areas in Malaysia: The Mediating Role of Social

Capital", otherwise:

·       Explain why Malaysia was selected as a case study.

·       Discuss the main findings with an international perspective.

Added in the title section:

“In Malaysia”

Income Disparity and Mental Wellbeing among Adults in Semi-urban and Rural Areas in Malaysia: The Mediating Role of Social Capital

1 (3)

2

The data was taken from previous survey in 2013. This should be discussed as one of limitations of the study. First, there might be changes in economic status in the case country a decade ago and now. Second, in the last two years, there are changes in terms of social interaction and mental health of respondents due to the pandemic, that might affect the association between income and mental illness. See for instance:

·       https://doi.org/10.3390/ijerph17217947

·       https://doi.org/10.3390/bs11050064

·       https://doi.org/10.3390/ijerph19063158  

Added:

“Finally, the study was conducted in 2013 and does not take into account the massive effect of the COVID-19 pandemic has had on mental illness among the Malaysian population. There are changes in terms of social interaction and mental health of respondents due to the pandemic, that might affect the association between income and mental illness [55].”

9 (445-449)

3

The findings should be discussed in a broader perspective, for instance, (economic and/or health) policy implications. 

Added in discussion:

“One explanation is that local initiatives such as PeKa B40, mySalam and household living aid (BSH) are continuously improved and disseminated among this group [42], more so than the M40 income group. Many of these programs are facilitated by local leaders and volunteers among these community. This provides a platform for continuous improvement of social capital within this population group.”

Note: PeKa B40, mySalam and household living aid (BSH) are economic and health aids provided to the B40 income group by the government.

8 (358-362)

4

The references must be improved by citing the most recent (2020 or newer) and relevant studies.

Added additional relevant references in discussion:

·       Rizal H, Said MA, Abdul Majid H, et al. Health-related quality of life of younger and older lower-income households in Malaysia. PLoS One. 2022;17(2):e0263751. doi:10.1371/journal.pone.0263751

·       de Miquel C, Domènech-Abella J, Felez-Nobrega M, et al. The Mental Health of Employees with Job Loss and Income Loss during the COVID-19 Pandemic: The Mediating Role of Perceived Financial Stress. Int J Environ Res Public Health. 2022;19(6):3158. doi:10.3390/ijerph19063158

8 (359)

9 (449)

5

Figures and tables should be self-explanatory and stand-alone. Define all acronyms and uncommon terms (e.g. M40, B40, RM3860,...)

Added defined terms in Table 1:

“B40 (Less than RM3860)

M40 (RM3860 to RM8319)”

6-7 (268-269)

6

Minor issues on capitalization and consistency in using comma for numbers (e.g. 25,512 respondents vs 6651).

Revised numbering comma:

“25512”

3 (129)

Round 2

Reviewer 1 Report

After evaluating the response of the authors on the comments of the manuscript, it is considered that they have been taken into

Author Response

Dear Reviewer,

Thank you for taking the time to review our manuscript and for your kind response.

Reviewer 2 Report

Dear authors, unfortunately, your answers were not fully compelling to me. 

  1. If you do not have enough participants in a specific DASS cluster/level, you could simply remove them from the analysis. But putting each type of Depression, Anxiety, and Stress from mild to above (thus including also severe) in the same basket is inappropriate to me. I am deeply unsatisfied with your answer even because, apart from not having adjusted the paper accounting for my point, you did not even discuss this issue properly as a main possible limitation. 
  2. Since there is uncompelling evidence about DASS validation in your country, I would request to mention possible measurement bias coming from this not fully validated measure. 
  3. I do not agree with the fact that the reanalysis of this model including the sex dimension would fall short of the scope of this study. Instead, is something that in my opinion you should do given the well-known relationship between sex and DASS. So, I am requesting a modification in such a direction. 

Author Response

Dear Reviewer,

Thank you very much for reviewing our manuscript and for your response. We apologize for not responding to your comments vigilantly and therefore we conducted a full re-analysis of the model as requested. We provide the following response:

1    If you do not have enough participants in a specific DASS cluster/level, you could simply remove them from the analysis. But putting each type of Depression, Anxiety, and Stress from mild to above (thus including also severe) in the same basket is inappropriate to me.    

Added as limitation in the discussion: 
“The dependent variable was dichotomized as those who do not experience depression, anxiety, and stress; and those who have mild, moderate, severe, and extremely severe depression, anxiety, and stress levels. In this way, it is not possible to distinguish between the different types of discomfort people experience.”    

9 (446-449)

2    Since there is uncompelling evidence about DASS validation in your country, I would request to mention possible measurement bias coming from this not fully validated measure.    

Added as bias limitation:
“Currently, there are no normative scores for DASS-21 in Malaysia, however, a number of validation studies were conducted but are not derived from representative populations [55-56] which may lead to measurement bias.”

55.    Ramli, M; Salmiah, M; Nurul, Ain, M. Validation and pychometric properties of Bahasa Malaysia version of the Depression Anxiety and Stress Scales (DASS) among diabetic patients. MJP Online Early. 2009, 8, 1-7.
56.    Musa, R; Ramli, R; Abdullah, K; Sarkarsi, R. Concurrent validity of the depression and anxiety components in the Bahasa Malaysia version of the Depression Anxiety and Stress scales (DASS). ASEAN J Psychiatry. 2011, 12, 66-70.

Additional references:
•    Osman ZJ, Mukhtar F, Hashim HA, Abdul Latiff L, Mohd Sidik S, Awang H, Ibrahim N, Abdul Rahman H, Ismail SI, Ibrahim F, Tajik E, Othman N. Testing comparison models of DASS-12 and its reliability among adolescents in Malaysia. Compr Psychiatry. 2014 Oct;55(7):1720-5. doi: 10.1016/j.comppsych.2014.04.011
•    Rusli BN, Amrina K, Trived S, Loh KP, Shashi M. Construct validity and internal consistency reliability of the Malay version of the 21-item depression anxiety stress scale (Malay-DASS-21) among male outpatient clinic attendees in Johor. Med J Malaysia. 2017 Oct;72(5):264-270.

9 (449-451)

3    It would be important to report descriptive statistics and results in a sex-sensitive way (i.e., disaggregating data and results by sex if applicable). Thus, it would be important to re-run the models and see if differences will emerge considering males and females separately. Indeed, sex, as the authors surely know, is strongly related to DASS variables. 

I do not agree with the fact that the reanalysis of this model including the sex dimension would fall short of the scope of this study. Instead, is something that in my opinion you should do given the well-known relationship between sex and DASS. So, I am requesting a modification in such a direction.  

Explored and re-run the model in a sex-sensitive way 
The univariate and the multivariable logistic with interaction terms was created to look for the effect of gender. However, none of the models showed any meaningful findings. Usually, we only proceed with stratification when our data showed a positive interaction. Therefore, the analysis showed gender is not an effect modifier.

Kindly refer to the log files for more detail on the full analysis.

Log added – supplementary files below via link

https://figshare.com/s/8c82958b284c47ddd0ca
